# The Effects of Over-Ground Robot-Assisted Gait Training for Children with Ataxic Cerebral Palsy: A Case Report

**DOI:** 10.3390/s21237875

**Published:** 2021-11-26

**Authors:** Myungeun Yoo, Jeong Hyeon Ahn, Eun Sook Park

**Affiliations:** Department and Research Institute of Rehabilitation Medicine, Severance Hospital, Yonsei University College of Medicine, Seoul 03722, Korea; YME8902@yuhs.ac (M.Y.); AJH1208@yuhs.ac (J.H.A.)

**Keywords:** over-ground, robotic training, gait, postural balance, ataxic, cerebral palsy

## Abstract

Poor balance and ataxic gait are major impediments to independent living in ataxic cerebral palsy (CP). Robot assisted-gait training (RAGT) has been shown to improve the postural balance and gait function in children with CP. However, there is no report on the application of RAGT for children with ataxic CP. Here, we report two cases of children with ataxic CP who underwent over-ground RAGT along with conventional therapy for 4 weeks. Outcome measures including the gross motor function measure (GMFM), pediatric balance scale, pediatric reach scale, one-minute walk test, and Timed Up and Go test were assessed before and after the 4-week intervention. Both cases were well adapted to the RAGT system without any significant adverse event. Improvements in the GMFM after RAGT, compared with that in the GMFM, after intensive conventional therapy have been reported previously. It is noteworthy that over-ground RAGT improved areas of the GMFM that did not improve with conventional therapy. In addition, over-ground RAGT with conventional therapy led to improvements in functional balance and walking capacity. These findings suggest that over-ground RAGT is feasible and may be a potential option for enhancing balance and functional walking capacity in children with ataxic CP.

## 1. Introduction

Cerebral palsy (CP) is the most common disorder affecting motor function in childhood. Ataxic CP is the least common form of CP, accounting for 2.5–5% of all CP cases [1,2]. Ataxic CP is characterized by abnormal patterns of posture or movement and loss of orderly muscular coordination, affecting the force, rhythm, and accuracy of the limbs and trunk, which leads to poor limb coordination, target accuracy, and balance [3]. These problems lead to difficulties in activities of daily living including poor postural balance and ataxic gait.

Gait limitation is one of main impairments in children with CP. According to the study by the Surveillance of Cerebral Palsy in Europe, 30% of children with CP have no walking ability at the age of 5 years [4]. The use of robot-assisted gait training (RAGT) to train gait in children with CP has increased over the last decade. Tethered RAGT such as Lokomat (Hocoma AG, Volketswil, Switzerland) has been used in children with CP [4,5]. This device uses a customized suspension system and exoskeleton to provide support over the treadmill. This tethered exoskeleton tends to be more stable and inherently safer than that of an untethered RAGT [6]. However, this system only moves the children according to a predetermined fixed trajectory or timing; thus, it can restrict patients’ active participation [7]. Previous studies have reported the benefits of tethered RAGT for children with CP. However, according to a systematic review, there is weak and inconsistent evidence supporting the benefits of tethered RAGT [5].

Recently, untethered exoskeletons, such as ReWalk (ReWalk Robotics Inc., Marlborough, MA, USA), Indego (Parker Hannefin Corp, Mayfield Heights, OH, USA), Hybrid Assistive Limb (HAL; Cyberdyne Inc, Tsukuba, Japan), and the Ekso (Ekso Bionics, Richmond, CA, USA), have been used for RAGT in patients with spinal cord injury or stroke [8,9,10]. Untethered exoskeletons are wearable, articulated suits with self-contained power sources and control algorithms that lead to a significant reduction in physical constraints on the patients, allowing them to move freely [6,11]. This system permits over-ground gait training. Alias et al. reported that over-ground training with exoskeletal support can potentially improve outcomes because its biomechanics are similar to that of the natural gait, and it assists patients with their daily routine [11].

Angel Legs M (ANGEL ROBOTICS Co., Ltd., Seoul, Korea) is an untethered wearable robot system with actuators in both the hip and knee joints. A previous study reported that the use of this robotic system was feasible and had the potential to enhance the gross motor function and walking capacity of children with spastic CP [12]. To the best of our knowledge, no study has reported the feasibility and benefits of untethered RAGT for children with ataxic CP. Here, we report the changes in gross motor function and postural balance in two children with ataxic CP after over-ground RAGT, which was well tolerated. This report may facilitate the use of this system for children with ataxic CP.

## 2. Case Description and Methods

### 2.1. Case Description

#### 2.1.1. Case 1

An 11-year-old girl reported to our clinic at the age of 2 years and presented with overall hypotonia and developmental delay. Brain MRI findings revealed cerebellar atrophy. At the age of 2 years, she could sit without arm support and cruise along the furniture but could not walk independently. She received physical and occupational therapies. At 26 months old, she could walk independently at a slow pace under protective supervision. After she turned 7 years old, she was admitted to our clinic once or twice per year for a 4-week intensive physical therapy. As a result, the child had been hospitalized six times from 7 to 11 years old. The gross motor function measure (GMFM) was scored before and after each intensive therapy. Changes in GMFM during the last three hospitalizations are shown in Table 1. During the 4th and 5th admissions, there were no changes in dimension D. According to a previous study, the minimal clinically important differences (MCIDs) for GMFM scores were reported [13]. This case showed MCIDs in dimension D in the 3rd admission and in dimension E in all the three admissions. In addition, the total scores of GMFM during the 3rd and 4th admissions showed MCIDs. This patient received the over-ground RAGT along with conventional therapy during the 6th admission.

#### 2.1.2. Case 2

A 12-year-old girl visited our clinic at the age of 3 years due to developmental delay, and she presented with overall hypotonia; she could not walk or stand independently. The developmental ages based on the Denver developmental screening tests were 12 months for the gross motor domain, 24 months for the fine motor and adaptive and personal–social domains, and 30 months for the language domain. A brain MRI finding revealed cerebellar atrophy. To undergo and receive intensive therapy, she was admitted to our clinic once or twice per year. Consequently, she was hospitalized 12 times and received intensive physical and occupational therapies for 4 weeks. Changes in GMFM with inpatient intensive therapy are shown in Table 2. There were no MCIDs in changes in dimensions D and E and total scores over the 9th–11th admissions. During the 12th admission, she underwent over-ground RAGT along with conventional therapy.

### 2.2. Methods

#### 2.2.1. Training Program

For the over-ground RAGT, we used Angel Legs M20, the untethered wearable torque-assisting exoskeleton robot, which assists joint torque generation according to the gait phase that is automatically detected by the ground contact sensors beneath the soles (Figure 1e). The joint actuators (Figure 1i,j) can generate both flexion and extension torque at the hip and knee joints. The actuators are controlled by a computer in a backpack (Figure 1h), such that the whole-body weight is partially supported during the stance phase by applying an extension torque at the hip and knee joints, and the weight of the leg is supported during the swing phase by applying a flexion torque at the hip and knee joints. To determine the appropriate amount of assistive joint torque, a spring-loaded inverted pendulum model of gait dynamics is used. The ankle joints are not actuated [12] but are mechanically limited to prevent hyper-dorsi/plantar flexion. The actuator modules consist of brushless direct-current motors (70 W, EC45-C45 a Maxon motor Ltd., Sachseln, Switzerland), customized gear sets, and sensors for measuring the motor angular position and the absolute angle of the human joint. The maximum magnitude of an assistive torque is approximately 18 Nm in a continuous assistance condition.

Over-ground RAGT was performed for 20 sessions (30 min per session, five sessions per week) as performed in a previous study [12]. All training was conducted by a physical therapist with over 10 years of experience in CP rehabilitation. The patients underwent the RAGT without using a walking aid. The same duration and intensity of physical therapy and occupational therapy, as provided during previous admissions, were provided during the over-ground RAGT period.

#### 2.2.2. Outcome Measures

The clinical assessments were performed before and after the RAGT. The GMFM, Pediatric Balance scale (PBS), Pediatric Reach Test (PRT), 1 Minute Walk Test (1MWT), and Timed Up and Go (TUG) test were assessed as therapeutic outcome measures.

The GMFM is a standardized outcome measure of gross motor function and widely used to assess the changes in gross motor function over time in children with CP [14]. It consists of five dimensions (A, lying and rolling; B, sitting; C, crawling and kneeling; D, standing; and E, walking, running, and jumping). The sum of the items of each dimension is recorded as a percentage [14].

The PBS is derived from the Berg Balance Scales for assessing postural control in children. PBS is a useful tool to assess the functional balance of children with CP, which can be used reliably in clinical practice to indicate the motor impairment level, from mild to moderate impairment, of children with CP [15]. The PBS score is an aggregation of the scores of 14 tasks and can range from 0 to 56. Higher scores indicate better postural control. The children were evaluated following the sequence proposed by the scale [15].

The PRT is a simple, valid, and reliable measure to assess balance by measuring the distance a child could reach forward, to the right, and to the left in both sitting and standing positions [16,17].

The TUG test measures the time, in seconds, taken by an individual to stand up from a standard armchair and walk a distance of 3 m, turn, walk back to the chair, and sit down again. It is a reliable and practical screening tool that measures basic functional mobility and balance [18,19,20].

The 1MWT is a valid and reliable tool used for children with CP. In this test, a physiotherapist instructs the child to walk as fast as possible around a 40 m circular track over a period of 1 min; running is not allowed. If a child falls or needed to take a break during the 1MWT, the measurement is considered incomplete [21]. It is considered as a test that represents everyday walking capacity more precisely than other tests in patients with CP patients by assessing how fast they can move [21].

## 3. Results

There were no major adverse events or safety issues during the over-ground RAGT, and both children were well adapted to the training without any complaints.

The changes that occurred after the RAGT are shown in Table 3. Both cases showed MCIDs in dimensions D and E and total scores of GMFM [13].

Postural balance, measured using the PRT and PBS, also improved after the RAGT (Table 4). The MCIDs in the PBS for children with CP was reported in a previous study [22]. Both cases showed MCIDs in the PBS. There has not yet been reports on MCIDs in PRT; minimal detectable change (MDC) for PRT was reported as 46.5 cm in a previous study [17]. Case 1, but not case 2, showed an MDC in PRT, and the changes in PRT in case 2 was approaching the MDC.

The changes in 1MWT and TUG tests are shown in Table 5. MCIDs in 1MWT and TUG tests in children with CP were reported in a previous study [20]. Both cases showed MCIDs in the 1MWT and TUG tests.

## 4. Discussion

RAGT with an untethered exoskeleton is an emerging field; thus, only a few trials of RAGT with an untethered exoskeleton have been conducted as case reports or single arm studies, showing improvements in the walking ability, gross motor function, and balance in patients with CP [12,23,24,25]. In addition, there is a cross-over randomized controlled study showing the changes in gait kinematics and kinetics, which occurred after only 5 min of RAGT with untethered exoskeleton [26].

The application of RAGT in children with ataxic CP has rarely been reported. Among the motor types of CP, the incidence of ataxic CP is very low [1,2], and even in population-based studies, the number of cases is very small. Therefore, the proportion of children with ataxic CP who have the ability to walk is varied across studies. According to a previous study, 10% of children with ataxic CP cannot walk [4], while 100% children walked with or without a walking aid in other studies [1,2]. Children with ataxic CP who can walk require constant close supervision to prevent unexpected falls due to poor walking balance. Poor walking balance is one of the major obstacles to independent living in children with ataxic CP.

In previous studies, the over-ground RAGT with a wearable robot system has been used in the treatment of patients with spinocerebellar ataxia [27,28]. They found that these patients could walk more smoothly with an increased harmonic ratio while walking with the wearable robot system than without the system. In addition, other studies have demonstrated improvements in gait function and postural balance after weeks of RAGT with the wearable robot system in patients with spinocerebellar ataxia [29,30]. These findings suggest the potential benefit of over-ground RAGT for patients with ataxia. The present study also demonstrated MCID (or MDC) in the PBS, PRT, 1MWT, and TUG tests after over-ground RAGT in children with ataxic CP. These improvements in postural balance and walking capacity in our study are consistent with that observed in previous studies. Unexpected frequent falls represent a major obstacle that prevent children with ataxic CP from walking and living independently. These improvements suggest that RAGT may be a hopeful therapeutic option for children with ataxia. However, both previous studies and this study are case studies; thus, the effectiveness of over-ground RAGT over conventional therapy remains uncertain. Further study is therefore warranted to address this uncertainty. Both tethered and untethered RAGT have demonstrated some improvements in walking ability, gross motor function, and postural balance [12,23,24,25,31,32,33]. However, the evidence to support the benefits of both tethered and untethered RAGT for children with CP is still weak. Moreover, there are few studies on the effect of either tethered or untethered RAGT for children with ataxic CP. The untethered RAGT provides task-specific over-ground training with the most freedom and realistic walking experience and assists patients with their daily routine [12]. Therefore, this robotic system may be more beneficial for enhancing the postural balance and walking capacity. To the best of our knowledge, no study has investigated the efficacy of untethered RAGT, compared with tethered RAGT, in children with CP. Further studies should clarify whether untethered RAGT is better than tethered RAGT for enhancing the postural balance and walking capacity.

In this study, the same intensity and duration of physical and occupational therapies were provided during all of the hospitalizations of the patients. Thus, comparing the changes in GMFM with RAGT during the last admission with the changes in GMFM with conventional therapy during previous admissions may reveal the benefits of RAGT over conventional therapy. In this study, case 1 showed MCIDs in dimensions D and E with both RAGT and conventional therapy only, while case 2 showed MCIDs with only RAGT. The scores of dimension D in case 1, which were stable over the 4th and 5th admission periods, changed remarkably with over-ground RAGT during the 6th admission. In case 2, the scores of dimensions D and E and total GMFM showed no further improvements during the last three hospitalizations, but the over-ground RAGT resulted in MCIDs in those scores. The improvements observed after over-ground RAGT in areas of the GMFM that did not improve during previous admissions support the potential benefits of over-ground RAGT on gross motor function in these children.

This case study has some limitations. Both patients underwent both conventional therapy and over-ground RAGT; thus, without a control group, it is unclear the extent to which over-ground RAGT contributed to the improvement observed in our cases. Further studies should include a control group. Since the prevalence of ataxic CP is very low, it is very difficult to obtain a sufficient number of children with ataxic CP for studies with high levels of evidence. Cross-over randomized controlled studies may be helpful to verify the effect of over-ground RAGT in these children. The robotic system used in this study provided the patients with freedom and realistic walking experience as reported in a previous study [6]; thus, we consider that this system could be better than tethered RAGT, especially for children with ataxic CP. Comparison of the effects of tethered and untethered RAGT on function and balance, which can be investigated in future studies, will be valuable and interesting to clinicians involved in the management of such children.

## 5. Conclusions

To the best of our knowledge, this is the first study to demonstrate the feasibility and benefits of over-ground RAGT in children with ataxic CP. Good adaptability to over-ground RAGT without major adverse events or safety issues suggests the feasibility of over-ground RAGT in children with ataxic CP. In addition, it is worth noting that over-ground RAGT can improve areas of the GMFM that do not improve with conventional therapies.

## Figures and Tables

**Figure 1 sensors-21-07875-f001:**
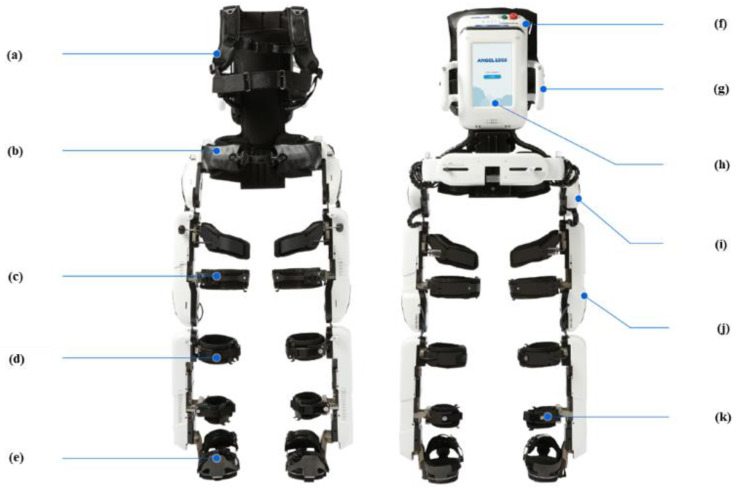
Untethered exoskeleton used in this study. (**a**) shoulder strap, (**b**) back strap, (**c**) thigh band, (**d**) calf band, (**e**) outsole, (**f**) battery cover, (**g**) handles held by an assistant to prevent side-to-side sway and assist weight-shift during training, (**h**) tablet used to control the starting or stopping of the robot, (**i**) actuator for the hip joint, (**j**) actuator for the knee joint, (**k**) ankle band.

**Table 1 sensors-21-07875-t001:** Changes in GMFM during the 3rd to 5th admissions for case 1.

		GMFM-88 (%) *			
		C	D	E	Total
3rd admission (8 July 2019)	Pre	97.62	79.49	41.67	83.76
	Post	97.62	84.62	55.56	87.56
4th admission (21 February 2020)	Pre	100	79.49	59.72	87.84
	Post	100	79.49	73.61	90.62
5th admission (12 January 2021)	Pre	100	79.49	73.61	90.62
	Post	100	79.49	77.78	91.45

* The GMFM (gross motor function measure); C, D, and E represent dimensions of GMFM. C, crawling and kneeling; D, standing; E, walking, running, and jumping.

**Table 2 sensors-21-07875-t002:** Changes in GMFM during the 9th to 11th admissions for case 2.

		GMFM-88 (%) *			
		C	D	E	Total
9th admission (14 August 2020)	Pre	92.86	79.49	58.33	86.14
	Post	92.86	79.49	59.72	86.41
10th admission (7 December 2020)	Pre	92.86	79.49	62.50	86.97
	Post	92.86	79.49	62.50	86.97
11th admission (8 March 2021)	Pre	97.62	87.18	50.00	86.96
	Post	97.62	87.18	50.00	86.96

* The GMFM (gross motor function measure); C, D, and E represent dimensions of GMFM. C, crawling and kneeling; D, standing; E, walking, running, and jumping.

**Table 3 sensors-21-07875-t003:** Changes in GMFM after RAGT intervention.

		GMFM-88 (%) *					
		A	B	C	D	E	Total
case 1	Pre	100	100	100	79.49	77.78	91.45
	Post	100	100	100	92.31	80.56	94.57
case 2	Pre	100	100	97.62	87.18	51.39	87.24
	Post	100	100	100	89.74	54.17	88.78

* The GMFM (gross motor function measure); A, lying and rolling; B, sitting; C, crawling and kneeling; D, standing; E, walking, running, and jumping.

**Table 4 sensors-21-07875-t004:** Changes in PBS and PRT after RAGT.

		Case 1	Case 2
PBS	Pre	35	35
	Post	45	42
PRT in standing			
Forward reach (cm)	Pre	21	13
	Post	35	25
Right reach (cm)	Pre	8	8
	Post	24	15
Left reach (cm)	Pre	10	12
	Post	26	14
PRT in sitting			
Forward reach (cm)	Pre	36	21
	Post	42	34
Right reach (cm)	Pre	16	13
	Post	32	20
Left reach (cm)	Pre	21	17
	Post	35	20
PRT total (cm)	Pre	112	84
	Post	194	128

PBS, Pediatric Balance Scale; PRT, Pediatric Reach Test.

**Table 5 sensors-21-07875-t005:** Changes in one-minute walk test and Timed Up and Go test after RAGT.

		Case 1	Case 2
1MWT (meter)	Pre	46.16	10.10
	Post	61.06	21.30
TUG test (s)	Pre	25.42	39.95
	Post	19.58	24.51

1MWT, 1 Minute Walk Test; TUG, Timed Up and Go.

## Data Availability

The data presented in this study are available on request from the corresponding author. The data are not publicly available due to reasons concerning privacy of the participants.

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
