# Peer review of "The Effects of Over-Ground Robot-Assisted Gait Training for Children with Ataxic Cerebral Palsy: A Case Report"

_sensors, 2021, doi:10.3390/s21237875_

Round 1

Reviewer 1 Report

This paper present the early study of the feasibility and benefits of over-ground RAGT in children with ataxic CP, which patient number is less and difficult to evaluate.

A comparison of various evaluation items before and after use suggests the possibility of the proposed RAGT, although there are two cases. However, as described in Limitation, it is also an important point for comparison with the tethered RAGTs, and it is desirable to describe more clearly what kind of effect this proposal type has, based on the outcome measures' results.

Overall well organized and well written as a case report.

Reviewer 2 Report

Respected Authors, 
Thank you very much for the opportunity for revieweing the manuscript entitleg "The effects of over-ground robot assistive gait training for chil- 2
dren with ataxic cerebral palsy: A case report "

Undoubtedly, the topic reflects to an important clinical challange. The rehabilitation process (both "conventional" and robot-assisted) is exteremely important in children with ataxic CP. This case-reports describes a method with over-ground exoskeleton (Angel Legs), involving 2 children. 

The abstract is well structured and informative. The introduction is detailed enough, and summarizes the most imporant previous results on the field. The protocol of the study is clear, however, a questions arises:
1.) Is it possible to to indicate the "timeline" of each hospital admission? It would be more informative, since it does not contain information how much time has been exactly passed between the admissions, which is an imporant aspect in terms of rehabilitation. 

The results are well presented, and internationally recognized and standardized methods have been used (eg. 1MWT, TUG etc.).  Here, I would like to ask/propose the followings: 

1.) During the tests, did the research group perfomed any movement analysis? The description of change in joint-movements would be interesting (e.g. change of the threshold of joint movement etc.)

2.) Is there any information regarding the engagement or satisfaction of the users, how  did they like the RAGT , comapred to conventional therapy? (E.g. using a Likert scale)

The other results are well-presented and the conclusions support the findings. Also, the limitations are correctly described. 

In my opinion, it is an interesting case report, introducing an emerging technology, with a special target group (children) and the novelty is clear. 
I would recommend it for acceptance after answerint for the questions (minor revision). 

Respectfully,
